# Compressed sensing expands the multiplexity of imaging mass cytometry

Tsuyoshi Hosogane [1,2,3,6], Leonor Schubert Santana[2,7], Nils Eling [1,2], Holger Moch [4,5] & Bernd Bodenmiller [1,2]

The multiplexity of current antibody-based imaging is limited by the number of reporters that can be detected simultaneously. Compressed sensing can be used to reconstruct high-dimensional information from low-dimensional measurements. Previously, compressed sensing using composite in situ imaging (CISI) of transcriptomic data leveraged gene co-regulation structure to recover spatial expression of 37 RNA species from images of 11 composite channels. Here, we extend the CISI framework to protein expression data measured by imaging mass cytometry (IMC). CISI-IMC accurately recovers spatial expression of 16 immune and stromal marker proteins from images of 8 composite channels with an average Pearson's correlation of 0.8 across protein. Training the CISI-IMC framework using data collected on multiple human tissues enables universal decompression of composite data from a wide range of tumor and healthy tissue types. The expression dictionary and barcoding matrix described here are immediately implementable for general immune and stromal cell type classification, but CISI-IMC can in principle be applied to other markers or other antibody-based imaging methods. Our work lays the foundation for much higher plex protein imaging.

Antibody-based protein imaging allows profiling of the spatial distribution of cell phenotypes across tissues, enabling the study of tissue function in health and disease. Detection of biomolecules in standard immunofluorescence imaging is limited by the number of available fluorescent channels, typically five, due to spectral overlap. This limitation has been overcome by performing iterative staining and de-staining using a few fluorescently labeled antibodies in each cycle, with each channel in each round of staining typically corresponding to a specific marker[1–6]. Iterative methods such as CODEX, 4i, and CyCIF can achieve up to 60-plex imaging[2,3,7]. Higher multiplexity has been achieved using combinatorial barcoding, where the sequence of channels over imaging rounds creates a unique barcode for each target molecule. For example, CosMx, which employs combinatorial barcoding, has been used to image 108 proteins in a tissue sample[8].

Limitations of these methods are autofluorescence and tissue damage caused by repeated staining cycles and increasing measurement time with each cycle. Combinatorial barcoding is also confounded by molecular crowding as this technique requires single-molecule detection of each target molecule for accurate decoding. Even higher multiplexity can be achieved by sequencing-based approaches, in which antibodies are labeled with a DNA tag that carries a barcode that can be sequenced to identify the location on the slide[9]. For example, 300-plex imaging was demonstrated with spatial CITE-seq[10]. Sequencing-based methods do not suffer from autofluorescence and do not require repeated staining cycles, but currently the resolution is too low (25 μm) to allow single-cell analyses.

In contrast to the fluorescence- and sequencing-based methods, mass spectrometry-based imaging methods such as multiplex ion

[1]Department of Quantitative Biomedicine, University of Zurich, Zurich, Switzerland. [2]Institute for Molecular Health Sciences, ETH Zurich, Zurich, Switzerland. [3]Life Science Zurich Graduate School, ETH Zurich and University of Zurich, Zurich, Switzerland. [4]Department of Pathology and Molecular Pathology, University Hospital Zurich, Zurich, Switzerland. [5]Faculty of Medicine, University of Zurich, Zurich, Switzerland. [6]Present address: Division of Molecular Neurobiology, Department of Medical Biochemistry and Biophysics, Karolinska Institute, Solna, Sweden. [7]Present address: Wolfson Wohl Cancer Research Centre, School of Cancer Sciences, University of Glasgow, Glasgow, Scotland, UK. ✉e-mail: bernd.bodenmiller@uzh.ch

beam imaging (MIBI) and imaging mass cytometry (IMC) enable over 40-plex imaging without sequential staining cycles at a spatial resolution of 1 μm or higher[11,12]. For both methods, antibodies are tagged with metal isotopes that are simultaneously detected by either secondary ion mass spectrometry (MIBI) or laser ablation and subsequent mass spectrometry (IMC). The multiplexity of MIBI and IMC is limited by the number of available channels to detect individual metal isotopes. MIBI and IMC already enable large scale sample analyses and since the number of measured antibodies does not affect measurement time, approaches to increase multiplexity are highly attractive to enable deeper tissue analysis. Combinatorial barcoding could be implemented by labeling each antibody with a unique combination of metal isotopes, but given the high abundance of proteins, single-molecule detection that is required for accurate decoding is not feasible for these techniques. Expanding MIBI and IMC beyond 40-plex would be possible with a method that can decode combinatorial barcodes without the need for single-molecule detection.

In the RNA imaging field, composite in situ imaging (CISI) has been implemented to decode combinatorial barcodes without single-molecule detection by using prior knowledge of transcriptome profiles[13,14]. Leveraging the biological principle of gene co-regulation, CISI constructs a dictionary of gene expression modules from training data, which enables the decoding into the spatial expression patterns of individual genes. By training on single-cell RNA sequencing data, CISI accurately recovered the spatial abundance of 37 RNA species from the combinatorially barcoded imaging data of only 11 channels[14]. CISI decoding can be performed on the single-cell or pixel-level and therefore only requires resolution high enough for single-cell analysis.

Here, we implement CISI for mass-tag based imaging approaches such as IMC and MIBI to expand multiplexity over the number of available isotope channels. We reason that a protein expression dictionary for IMC data could be constructed since proteins, like mRNAs, show inter-dependencies due to co-regulation. We perform 16-plex standard IMC imaging of multiple tissues using antibodies that mainly identify immune and stromal cells for training data and constructed a dictionary of protein expression patterns. We then implement combinatorial barcoding by mixing 16 antibodies each labeled with unique combination of 8 metal isotopes. We demonstrate that the obtained 8-channel CISI-IMC images for 71 different tissues are accurately decoded back to 16-plex spatial single-cell protein expression data (average Pearson's correlation of 0.8) using our optimized CISI algorithm.

## Results

### CISI-IMC principles

Our goal was to compress high-dimensional imaging data into low-dimensional composite images and then recover the original images using computational methods and prior knowledge of the patterns in imaging data (Fig. 1). To obtain low-dimensional, single-cell composite data using IMC, antibodies targeting $p$ proteins are labeled with unique combinations of $m$ isotopes, where $m < p$ (Fig. 2a). The barcoding matrix **A** describes the combination of isotopes labeled for each antibody. The IMC data obtained from a tissue stained with the barcoded antibodies to $p$ proteins is then linearly combined and compressed into single-cell composite data with $m$ channels (Fig. 2a). The matrix **X** contains single-cell protein expression profiles, and the matrix **Y** describes the measured single-cell intensity profiles of composite isotope channels. The task of decompression is to recover **X** (i.e., single-cell protein expression) from the measured matrix **Y** (i.e., measured intensities of composite channels).

The desired matrix **X** can be decomposed into two matrices (Fig. 2a). One is the dictionary of protein-expression modules **U** that summarizes prior knowledge of single-cell protein expression of a training dataset. Each protein-expression module is a pattern of protein expression that was observed in the training dataset, and the dictionary stores all the protein-expression modules. The second matrix is the single-cell module activity **W**, which describes how known modules are expressed in single cells. For each cell, a few modules are selected from the dictionary and linearly combined to approximate the single-cell protein expression. Experimental single-cell composite data **Y** are decompressed back to individual protein expression data by decomposing **Y** into the barcoding matrix **A**, the dictionary **U**, and single-cell module activity **W** (Fig. 2a). Since the barcoding matrix **A** and the dictionary **U** are known, we can estimate **W**.

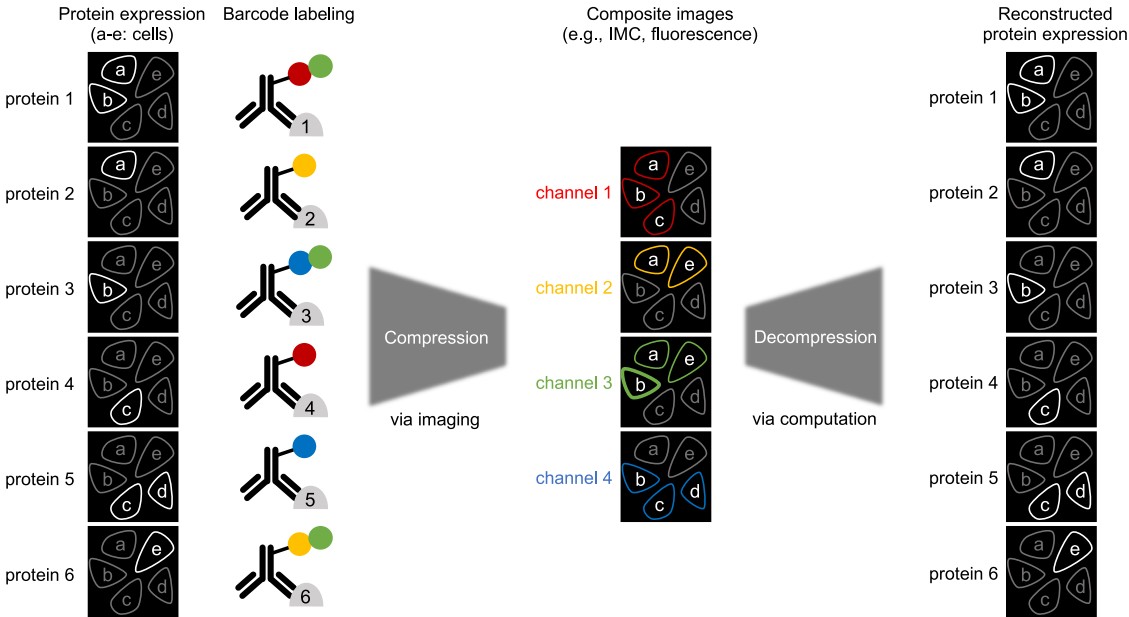

**Fig. 1 | Schematic of compressed sensing for antibody-based imaging.** Cells (a-e) expressing 6 proteins can be imaged in only 4 channels by labeling each antibody with a specific combination of channel reporters. Compressed sensing recovers the individual protein expressions from the composite images with an aid of prior knowledge about how the protein expressions are co-regulated.

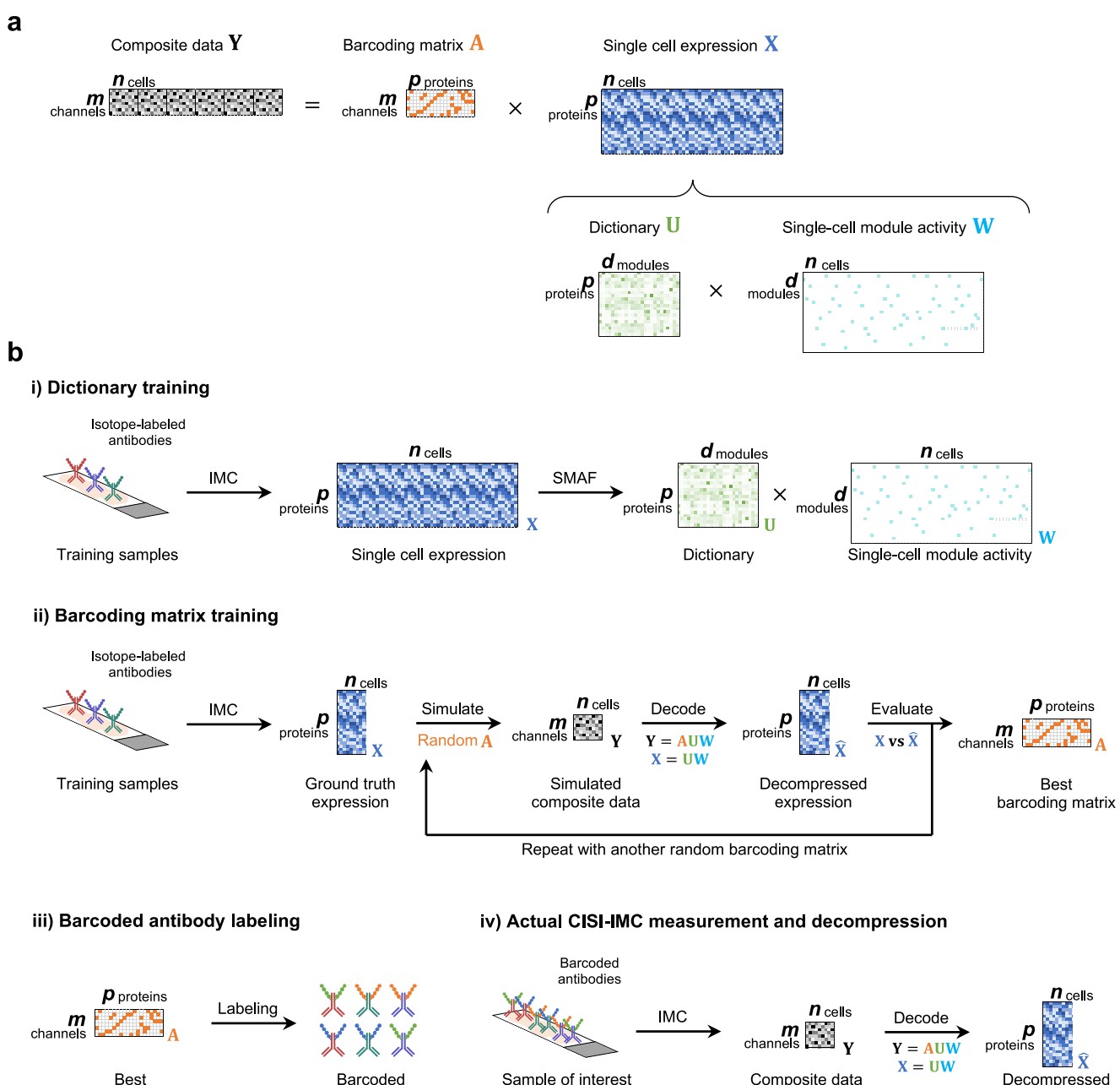

**Fig. 2 | CISI-IMC workflow. a** Schematic of the compression and decompression process using CISI-IMC. The barcoding matrix encodes the scheme for barcoding of *p* antibodies with *m* unique composite channels. By selecting an *m* smaller than *p*, single-cell protein expression data of *p* proteins can be compressed into single-cell composite data of *m* composite channels. To enable the decompression, single-cell protein expression data of *p* proteins is decomposed into a dictionary of *d* protein-expression modules and single-cell module activity with sparse activation of the modules for each cell. Decompression is performed by estimating the sparse single-cell module activity using the known barcoding matrix and the dictionary. **b** CISI-IMC workflow. During the CISI-IMC experimental workflow, a single-cell IMC data from a training dataset and the SMAF algorithm are first used to generate the dictionary. Second, the barcoding matrix is trained on a portion of the training dataset by simulating the single-cell composite data with randomly generated barcoding matrices, and simulated single-cell composite data are then decompressed back to single-cell protein expression data, which are compared to the original single-cell protein expression data. The barcoding matrix with the best decompressing performance is selected for the next steps. Third, antibodies are labeled with the specific combinations of metal isotopes according to the selected barcoding matrix. Finally, tissues of interest are stained with the barcoded antibodies and imaged with IMC. Obtained composite images are segmented into single-cell composite data that are decompressed back to single-cell protein expression data using the pre-trained dictionary.

## CISI-IMC workflow

The CISI-IMC workflow consists of four major steps: dictionary training, barcoding matrix training, barcoded antibody labeling, and CISI-IMC measurement and decompression into single-cell protein expression (Fig. 2b). To obtain a dictionary that efficiently stores protein-expression modules, we used a sparse module activity factorization (SMAF) algorithm on a training set of single-cell protein expression data from IMC measurements (Fig. 2b(i)). SMAF can efficiently summarize protein expression patterns into a dictionary of protein-expression modules by iteratively decomposing the training single-cell protein expression data into a dictionary and single-cell module activity while enforcing sparsity on both matrices, meaning each module in the dictionary has a few active proteins, and each cell has a few active modules.

Next, to design our barcoding matrix i.e., to select the optimal combination of markers compressed into each composite channel in the matrix, we simulated decompression performances of 8000 randomly generated barcoding matrices (Fig. 2b(ii)). For each such random matrix, we simulated single-cell composite data from the training dataset, decompressed it back into single-cell protein expression data, and compared it against the original single-cell protein expression data (i.e., the measured training data used for the simulation). We then selected the barcoding matrix with the best simulated decompression performance. Antibodies were then labeled with metal isotopes according to this selected barcoding matrix, tissues of interest were stained with the barcoded antibody mix, and then imaged with IMC (Fig. 2b(iii)). Composite images were segmented into single-cell composite data, which were decompressed into single-cell protein expression data using the pre-trained dictionary and barcoding matrix (Fig. 2b(iv)).

### Calculation of protein expression dictionaries using SMAF

We first confirmed that the SMAF algorithm from the CISI framework[14] can be applied to protein expression data generated by IMC. We chose 16 protein markers that define various cell types including immune and stromal cells that can be broadly found in various healthy and diseased tissues (Supplementary Data 1). We produced a training set of single-cell protein expression data $X$ by measuring protein expression with IMC on six tumor types (breast lobular carcinoma, breast ductal carcinoma, lung adenocarcinoma, lung squamous cell carcinoma, colon adenocarcinoma) and three healthy tissues (tonsil, appendix, lung). Tissue types were selected to include all the target cell types, and we assumed that the expression levels and patterns of cell type definition markers are stable across tissues.

SMAF was used to calculate a dictionary of protein expression modules $U$ by iteratively decomposing single-cell protein expression data $X$ into $U$ and a single-cell module activity matrix $W$. We tested several approaches for the application of SMAF to protein expression data ("Methods", Supplementary Figs. 1, 2). We found that, by modifying the parameters of the underlying algorithms, we could use SMAF to calculate protein expression dictionaries with different properties (e.g., number of protein-expression modules, sparsity), which could then be optimized based on simulated decompression performance (Methods).

### Optimization of the CISI-IMC training protocol

Next, to understand the effect of a barcoding matrix $A$'s property on the decompression performance, we simulated the single-cell composite data $Y$ using randomly generated $A$s on the single-cell training data $X$. The obtained $Y$ was decompressed back into $X$. Decompression performance was evaluated for each protein by calculating the Pearson's correlation coefficient between the decompressed $X$ and the original $X$. Each protein in a randomly generated $A$ was restricted to have up to two composite channel entries to minimize the complexity during the antibody labeling. Since the number of non-zero composite channels per protein in $A$ is equivalent to how many different isotopes are used to label a certain antibody, the imposed restriction guarantees that each antibody was labeled with no more than two isotopes. We observed that denser $A$ generally resulted in better decompression performance and that increasing the maximum composite channels per protein to 3 or 4 did not improve the performance for the tested compression rate of 16 proteins into 8 composite channels (Supplementary Fig. 3). Therefore, we used no more than two isotope labels per protein, and the number of proteins with only one composite channel in randomly generated barcoding matrices was restricted to no more than four to enforce the denser $A$.

Since the single-cell expression values in $X$ when calculating $U$ via SMAF are normalized (i.e., each protein in $X$ was scaled to the same vector size) but raw protein expression data are not normalized, we set

out to evaluate how the variations of norms in $X$ will affect decompression. Indeed, normalizing proteins in $X$ improved the simulated decompression performance (Supplementary Fig. 4a, b). Normalizing proteins in $X$ in CISI-IMC experimental data is not straightforward, however. As there are cell populations that do and do not express each protein marker, the norm of $X$ is affected by both the signal intensity and the density of cell population positive for the marker. In other words, a protein expressed only on a rare cell population would need to have higher signal intensity than a protein expressed on a common cell type in order to accurately normalize proteins in $X$. Since cell type abundance is dependent on the tissue sampled, and is difficult to account for without compromising the general applicability of the method, we aimed to achieve uniform signal intensity for each marker-positive cell population based on antibody titration tests. We prepared another training dataset $X$ using optimized antibody concentrations on the same tissues as used in the original training dataset. Simulating decompression using this training data indeed showed improved performance (Supplementary Fig. 4c, d). Therefore, titration of antibody or other detection reagents towards uniform intensity for each marker-positive cell population is recommended for barcoded labeling of target molecules.

### Barcoding matrix and dictionary for CISI-IMC

Next, we calculated the dictionary of protein expression modules for use in our CISI-IMC experiments. To finalize the dictionary $U$, we calculated $U$s with 12 different SMAF parameter conditions on 75% of the training data using optimally titrated antibodies and simulated $Y$. On the rest of the training data, we evaluated decompression performance using 200 randomly generated $A$s (Supplementary Fig. 5a). In addition, performance stability across tissues was assessed by evaluating the decompression performance for each tissue using a $U$ calculated on the training data, but excluding the data from the tissue of interest (Supplementary Fig. 5b). The final SMAF parameters were selected based on these experiments and on the performance stability across different normalization weights for training data, which simulated the variability in antibody concentration (Supplementary Fig. 5c). The final $U$ was obtained using finalized SMAF parameters (algorithm_for_W : Lasso, ldaU : 0.02, ldaW : 0.02, Num_blocks_W : 1) (Supplementary Fig. 5d). Next, to finalize the barcoding matrix $A$, we generated random $A$s barcoding 16 proteins in 8 composite channels, with each protein maximally using 2 channels. We simulated $Y$ and observed decompression performance using 2,000 randomly generated $A$s on 25% of training data using $U$ calculated with the finalized SMAF parameters on the rest of the training data. We performed the simulation four times and selected the top 50 $A$s from each simulation based on the Pearson's correlation for the worst performing protein, which resulted in 200 candidates from 8000 $A$s tested. Finally, we performed another set of simulations fixing the 200 $A$ candidates and selected the best $A$ based on the highest minimum protein correlation averaged over the four simulations. For the best $A$, the simulated average protein correlation coefficient was 0.924 and the minimum protein correlation coefficient was 0.864. We also tested the $A$s with 7 and 9 composite channels. As expected, increasing the number of composite channels improved the decoding results (Supplementary Fig. 6a). We selected 8 composite channels to balance performance and efficiency of the decompression (Fig. 3a). In summary, to perform CISI-IMC of 16 markers into 8 channels, we finalized the dictionary $U$ and barcoding matrix $A$ based on the best simulated decompression performance.

### CISI-IMC with 16 proteins compressed into 8 channels

Having finalized the dictionary and barcoding matrix, we moved on to perform CISI-IMC on target tissues. First, barcoded antibodies compressing 16 proteins into 8 composite channels were labeled according to the finalized barcoding matrix $A$. All unique antibody-isotope pairs

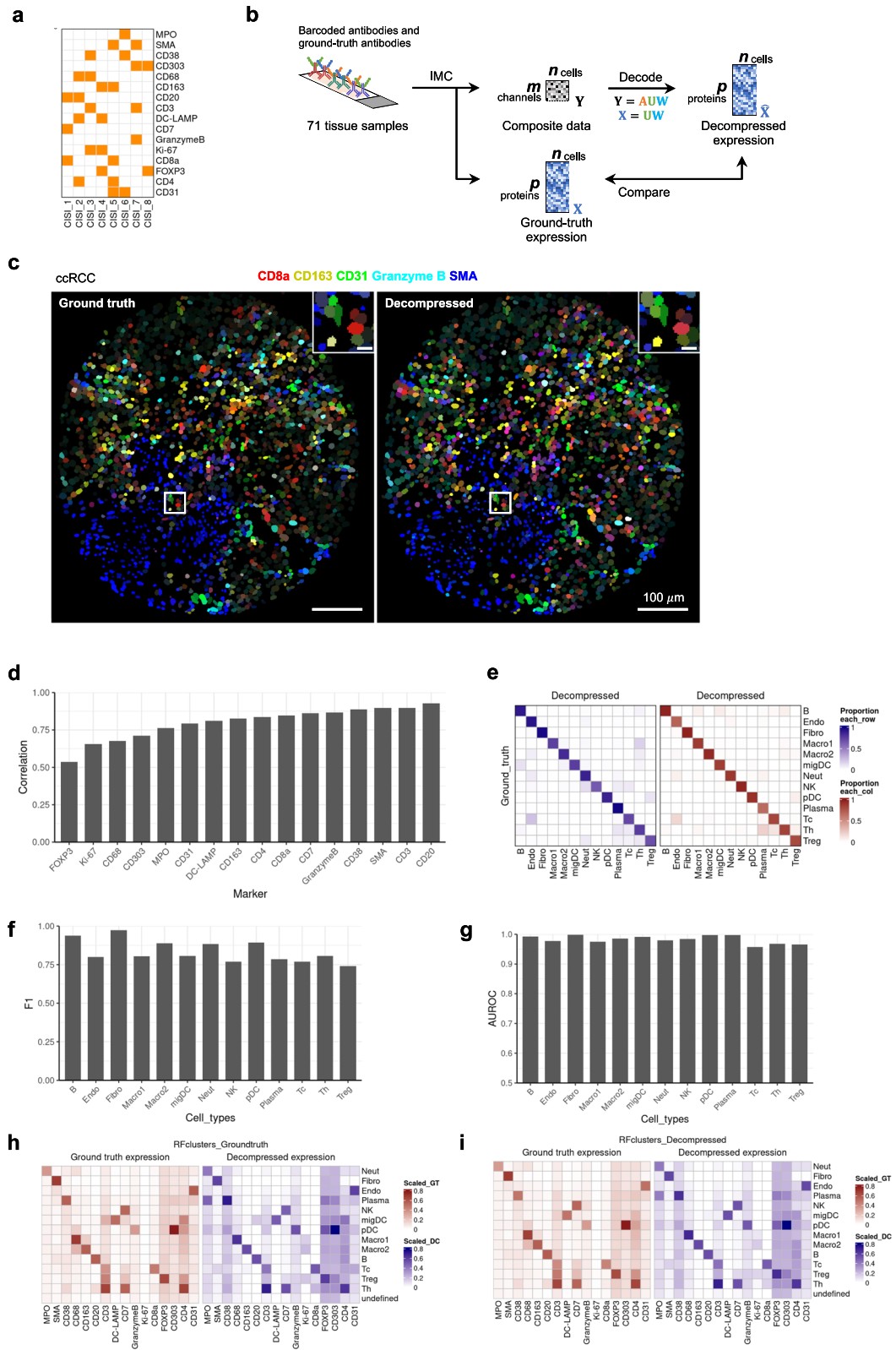

were titrated separately to identify optimal concentrations and were pooled together as a barcoded antibody mix. To evaluate the decompression performance of CISI-IMC experimentally, we stained 57 tumor tissue samples and 14 healthy or benign samples from various organs with the barcoded antibody mix. Our test samples included tissues with tertiary lymphoid-like structures or healthy germinal centers, and thus represented various immune cell types and densities.

In addition to the barcoded antibody-mix, we used a ground-truth antibody mix in which each antibody was conjugated to a single isotope (Fig. 3b). Hence, our evaluation dataset of 71 tissues contained single-cell composite data **Y** with matching ground-truth single-cell protein expression data **X**. The ground-truth **X** was also used to produce a simulated **Y**, which was compared against measured **Y** to assess the accuracy of the simulation performed to train the protein-

**Fig. 3 | Performance of CISI-IMC for 16-plex proteins in 8-plex composite channels on 71 tissues. a** Barcoding matrix describing the antibody labeling scheme of compressing 16 protein markers into 8 composite channels. **b** Schematic of performing CISI-IMC on 71 tissues samples along with measuring matched ground-truth protein expression data. Expression data from decompressing composite images were compared against ground-truth data to evaluate the performance of CISI-IMC. **c** Example images of ground-truth cell image and decompressed cell image from clear cell renal cell carcinoma. Displayed protein markers are indicated on top of the images. Scale bars are 100 μm. Top-right corner of each image is a magnified view of a region indicated by a white box. Scale bars are 10 μm in the magnified regions. **d** Pearson correlations between decompressed and ground-truth single-cell expression data for each protein. **e** Confusion matrix of cell-type annotation based on ground-truth data versus that based on decompressed data normalized row-wise (left) or column-wise. (right). B: B-cells, Endo: endothelial cells, Fibro: fibroblasts, Macro1: CD68+ macrophages, Macro2: CD163+ macrophages, migDC: migratory dendritic cells, Neut: neutrophils, NK: natural killer cells, pDC: plasmacytoid dendritic cells, Plasma: plasma cells, Tc: cytotoxic T-cells, Th: helper T-cells, Treg: regulatory T-cells. **f** F1 score for each annotated cell type. **g** Area under the receiver operating characteristic curve (AUROC) for each cell type. Median scaled protein expression values for each cell type assigned using the random forest classifier trained with **h** ground truth or **i** decompressed data.

expression dictionary **U** and the barcoding matrix **A**. For the decompression of the evaluation dataset, we used the dictionary **U** calculated from the training dataset, because we hypothesized that **U** from a diverse training dataset would be sufficient for the decompression of data from different tissues provided that the expression patterns for the selected 16 cell-type protein markers were stable across tissues.

CISI-IMC decompression accurately recovered individual protein expression data with an average Pearson's correlation of 0.8 between decompressed $\hat{X}$ and ground-truth **X**. The minimum correlation coefficient was 0.54 (Fig. 3c, d). The dictionary **U** calculated from training dataset was applicable to tissues not used in the training dataset, since decompression performance on the evaluation dataset did not improve by using **U** calculated from the evaluation dataset (Supplementary Fig. 7a, b). This experiment also confirmed that the simulation of composite data **Y** was accurate, since the decompression performance did not improve significantly when simulated **Y** was used instead of actual **Y** (Supplementary Fig. 7a, b). In addition, simulated **Y** and actual **Y** were highly correlated across all channels (Supplementary Fig. 7c).

To evaluate the extent to which errors in decompression influence downstream analyses, we classified cells into common cell types (i.e., B cells, T helper cells, and macrophages) using a random forest classifier. We trained classifiers for ground-truth **X** and for decompressed $\hat{X}$ separately, using known marker expression patterns for each cell type. After predicting cell types using ground-truth **X**, cells with low probability to be any cell type were excluded from the analysis on decompressed $\hat{X}$, as they were likely to be tumor cells or other cells that have low levels of expression of proteins detected with our antibody mix (see "Discussion"). We then assessed whether the decompressed $\hat{X}$ accurately classified cell types by comparing the identified cell types using decompressed $\hat{X}$ against those identified using the ground-truth **X** (Fig. 3e). All cell types were accurately recovered using decompressed $\hat{X}$ with a mean F1 score of 0.835 (Fig. 3f) and mean area under the receiver operating characteristic (AUROC) of 0.982 (Fig. 3g). Some cell types (e.g., B-cells and fibroblasts) showed excellent F1 score above 0.9. The decompressed $\hat{X}$ also accurately retained the unique protein expression pattern for each cell type (Fig. 3h, i). Hence, the error in the decompression for each protein expression had minimal impact on the downstream cell classification task. We conclude that CISI-IMC can be used across various tissues for compressing general cell type markers by 2-fold and retaining high cell type classification accuracy.

## Discussion

Here we demonstrated that CISI can extend the multiplexity of IMC beyond the number of available isotope channels. The SMAF algorithm was used to create a dictionary of protein expression modules based on a training dataset, and single-cell expression data for 16 proteins were accurately recovered from the measurement of 8 composite channels. With the compression ratio demonstrated here, up to 80 protein markers could in principle be compressed into the 40 isotope channels currently available in IMC. We also demonstrated that use of a training dataset based on images of different tumor and healthy tissues resulted in a dictionary that could be applied to accurately decompress a wide range of tissue types.

For our evaluation dataset of 71 tissues, the average Pearson's correlation for protein expression between decompressed $\hat{X}$ and ground-truth **X** was 0.80. The minimum correlation coefficient was 0.54 for the protein FoxP3, a marker for regulatory T cells that was rarely observed compared to other cell types in our dataset. Similarly, in previous applications of CISI to RNA imaging, transcripts that are rarely expressed and low-abundance transcripts were less accurately recovered[14]. In addition, background signal of FoxP3 staining was high (Fig. 3h, i), which may have also contributed to the reduction of decompression accuracy. In this work, we did not attempt to increase the signal intensity for rarely expressed and low-abundance proteins by increasing antibody concentration, since rarely expressed proteins differ depending on tissue origin and this could therefore limit the generalizability of our panel. Rather, we selected antibody concentrations that resulted in similar signal intensities for cells positive for each protein. Accordingly, we recommend that users titrate antibodies to similar signal intensities across markers when building a new antibody panel for CISI-IMC for use across multiple tissue types. In case a CISI-IMC antibody panel includes a marker for cells that are expected to be substantially more abundant than other cells in the panel, titrating such a marker to lower intensity than other markers may improve the overall decompression accuracy. In addition, incorporating multiplexed signal amplification approach would be useful to uniformly improve signal to background ratio. Increasing the number of composite channels resulted in improved decompression performance in our simulation. Therefore, depending on the accuracy of decompression required for a project, the compression ratio can be adjusted.

CISI-IMC is best used for cell type classification, rather than for quantification of marker expression. Cell type classification using decompressed protein expression was accurate with average F1 score of 0.85. Although this means some cells will be mis-annotated, it should be noted that cell type annotation using even standard IMC data typically includes some error due to signal spillover from imperfect cell segmentation. Our 16-plex antibody panel focused on immune and stromal cell markers. Other cell types such as tumor cells were not labeled and therefore not included in the cell type classification analysis. Indeed, we removed unlabeled cells from the analysis, for the purpose of removing tumor cells. This has the limitation that it may also have removed other ambiguous cell types not detected by our antibody panel, and we could therefore not assess decompression performance for such cells. In an experimental setting, we recommend that users include antibodies targeting all general cell types likely to be present, to avoid the scenario of a substantial number of unlabeled cells. Such markers may be included in the non-compressed channels. Further, markers for which quantification is critical should also be included in the non-compressed channel.

The expression dictionary and barcoding matrix we describe here is immediately implementable by IMC users for general immune and stromal cell type classification. Extension of the compression approach to other cell types is also possible by preparing a custom training dataset including antibodies for the cell types of interest, calculating a new dictionary with the recommended SMAF parameters (Methods), and selecting the barcoding matrix with the best simulated

decompression performance. Calculation of the dictionary and simulating random barcoding matrices should take about a day in total. Although we have only used IMC data for calculating the protein expression dictionary, other types of protein expression measurements, such as MIBI, iterative immunofluorescence imaging and also suspension mass cytometry, could be used for this purpose, although it may require overcoming unexpected technical issues. Similarly, a trained protein expression dictionary could in principle be used to decompress data from any protein measurement modality.

We reason that cell-level decompression will be sufficient for most multiplexed IMC protein analyses, since standard IMC data are almost always aggregated within each cell mask, and spatial analyses are performed on the single-cell level. In previous work, pixel-level decompression using deep learning models was demonstrated[14,15], and this strategy could theoretically be implemented for IMC. In cases where subcellular localization of protein expression (e.g., organelles, dendrites) is of interest, pixel-level decompression would be beneficial.

In summary, CISI-IMC is a highly multiplexed imaging approach that will allow expansion of IMC to 80-plex or higher. Current multiplexed protein imaging techniques with single-cell resolution can achieve ~60-plex imaging, but all rely on iterative imaging cycles, which requires complex image processing steps. IMC and MIBI are the exceptions that do not require iterative imaging cycles, but their multiplexities are limited to about 40 plex due to the number of available isotope channels. CISI-IMC retains the non-cyclic nature of IMC but expands the multiplexity beyond the available number of isotope channels.

## Methods

### Human tumor samples for training and evaluation datasets

The training dataset was derived from six human tumor tissues (lung adenocarcinoma, lung squamous cell carcinoma, colon adenocarcinoma, invasive lobular breast carcinoma, and two breast cancer) and three human healthy tissues (tonsil, lung, and appendix). The evaluation dataset was derived from tissue microarray composed of 71 tissues. Tissue of origin, tumor type, and ROI size for each tissue is described in Supplementary Data 2. The formalin-fixed, paraffin-embedded (FFPE) sections used were prepared at University Hospital Zurich. All the FFPE sections were kept at room temperature for short-term storage or at −20 °C for long-term storage. Use of samples received from University Hospital Zurich was approved by the Ethikkommission Kanton Zürich (KEK-ZH-Nr. 2014-0604, 100TO).

### Antibody labeling with metal isotopes

Isotope-labeled antibodies were used for IMC staining of training samples and evaluation samples. Antibodies were labeled with an isotope using the MaxPar X8 Antibody labeling kit (Fluidigm) according to the protocol supplied by the manufacturer. First, chelation was completed by incubating MaxPar X8 polymer, which carries terminal maleimide functionality and multiple chelators for lanthanide ions, in 2.5 mM lanthanide chloride solution (Fluidigm) at 37 °C for 30 min. The product was purified into C-buffer, provided with the MaxPar X8 Antibody labeling kit, using 0.5-ml, 3-kDa Amicon Ultra Filters (Millipore). In parallel, antibody was partially reduced in 0.8 mM TCEP at 37 °C for 30 min and was purified into C-buffer using 0.5-ml, 50-kDa Amicon Ultra Filters. Isotope-loaded polymer was added into partially reduced antibody and was incubated at 37 °C for 90 min. Conjugated product was purified over 0.5-ml, 50-kDa Amicon Ultra Filters. For CISI-IMC, each pair of isotope and antibody was separately conjugated and purified. Antibodies with different isotope labels were combined to produce the barcoded antibody mix.

### IMC protocol

FFPE sections were left at room temperature for 15 min after removal from storage. Deparaffinization was carried out using an AS-2 (Pathisto). The slides were placed into HIER buffer and heated at 95 °C for 30 min in a decloaking chamber (BioCare Medical) for epitope retrieval. Slides were cooled at room temperature for 20 min, washed in PBS for 15 min, and regions of interest were outlined with a hydrophobic pen (Vector Laboratories). Samples were then incubated with blocking buffer for 1 h at room temperature in a humidified chamber. Isotope-labeled antibody diluent was prepared in blocking buffer and incubated for overnight at 4 °C. Details of antibodies and their isotope labels are in Supplementary Data 1. Samples were washed in TBS for 15 min and incubated with 1:1000 dilution of 500 μM MaxPar Intercalator-Ir (Fluidigm) in PBS for 5–10 min, followed by a 15-min wash in TBS at room temperature. Slides were then dipped into deionized water for a few seconds, dried immediately using pressured air flow, and stored at room temperature until measurements. IMC images were acquired using a Hyperion Imaging System (Fluidigm). Laser ablation frequency was at 200 Hz, and pre-processing of the raw data into mcd files was completed using commercially available acquisition software (Fluidigm). Automated tuning of the argon flow and helium flow was performed on daily basis using a tuning slide coated with isotope-containing polymer (Fludigm).

### Data processing of IMC images into single-cell expression data

Steinbock v0.16.0 (https://github.com/BodenmillerGroup/steinbock) was used to convert the ion count raw data obtained from IMC software into single-cell expression data[16]. Briefly, DeepCell was used to segment the IMC images into single-cell masks. Aggregated images of cytoplasmic markers were used as cytoplasmic input, and iridium images were used as nuclei input. For CISI-IMC images, aggregated images of composite channels with cytoplasmic makers were used as cytoplasmic input. Single-cell expression data were obtained by calculating mean signal intensity for each channel in each cellular region defined by the cell mask. For the evaluation dataset, a custom hot pixel filtering method was used instead of the default hot pixel filtering method in Steinbock. The default method clips the hot pixel value to the maximum value of the surrounding 8 pixels, whereas the custom hot pixel filtering takes the minimum value of surrounding 8 pixels and propagates the value to the surrounding 8 pixels and self.

### SMAF algorithm

The dictionaries of protein expression modules were constructed from the training dataset using the SMAF algorithm. SMAF creates the dictionary by decomposing the training single-cell protein expression data ($\mathbf{X} \in \mathbb{R}^{p \times n}$) into a dictionary of protein expression modules ($\mathbf{U} \in \mathbb{R}^{p \times d}$) and single-cell module activity ($\mathbf{W} \in \mathbb{R}^{d \times n}$), where $p$ is the number of proteins, $n$ is the number of cells, and $d$ is the number of modules in the dictionary. The steps of SMAF are as follows: (1) Initialize $\mathbf{U}$ and $\mathbf{W}$ by non-negative matrix factorization. The initial number of modules $d_0 = 80$ was used for this work. (2) Fix $\mathbf{U}$ and calculate a sparse solution for $\mathbf{W}$ using Lasso or OMP. When using Lasso, the error tolerance coefficient $ldaW$ is provided to calculate the sparsest $\mathbf{W}$ within the error tolerance defined by $ldaW(=\lambda_W)$, where $\min_\mathbf{W}||\mathbf{W}||_1 s.t.||\mathbf{X}-\mathbf{UW}||_2^2 < \lambda_W||\mathbf{X}||_2^2$. When using OMP, the sparsity $k$ is provided to calculate the best fit solution, and up to $k$ modules can be active (non-zero) for each cell $u_n$ in $\mathbf{U}$ where $\min_\mathbf{U}||\mathbf{X} - \mathbf{UW}||_2^2 s.t. \forall n : ||u_n||_0 \le k$. (2) Fix $\mathbf{W}$ and calculate the sparse solution for $\mathbf{U}$ using Lasso with the error tolerance coefficient of $ldaU(=\lambda_U)$ where $\min_\mathbf{U}||\mathbf{U}||_1 s.t.||\mathbf{X}-\mathbf{UW}||_2^2 < \lambda_U||\mathbf{X}||_2^2$. (3) Perform module-wise L2 normalization of $\mathbf{U}$ (i.e., scale each module in $\mathbf{U}$ to the same vector size of 1). (4) Repeat steps (1), (2), and (3). In this work, 100 iterations were used.

### SMAF algorithm optimization

To enhance the decompression performance, SMAF enforces the sparsity in both $\mathbf{U}$ and $\mathbf{W}$ during the iteration. However, it is mathematically expected that sparser $\mathbf{U}$ and $\mathbf{W}$ lead to less accurate decomposition (i.e., the distance between $\mathbf{X}$ and $\mathbf{UW}$) (Supplementary Fig. 1a). We tested two

methods, least absolute shrinkage and selection operator (Lasso) and orthogonal matching pursuit (OMP), to balance sparsity and decomposition accuracy in calculating sparse solution for **U** and **W** during the SMAF iteration. First, we confirmed that 100 iterations were sufficient to obtain stable solutions for most of the conditions tested (Supplementary Fig. 1b, c), with varying levels of sparsity for **U** and **W** and decompression accuracy (Supplementary 1d–k). Next, we analyzed the dictionaries when using OMP or Lasso for calculating **W** during the iteration. For OMP, different sparsities $k$ (i.e., up to $k$ modules per cell are non-zero) were tested as OMP finds the best fit solution within the sparsity. For Lasso, different error tolerance coefficients ldaW were tested, as Lasso calculates the sparsest solution within the error tolerance. We used Lasso with different ldaU for calculating **U** during the iteration. Sparsities of the decomposed matrices and error tolerance showed a trade-off relationship as expected, regardless of using OMP or Lasso for calculating **W**. A looser error tolerance coefficient for calculating **U** (i.e., larger ldaU) was able to produce sparser **U** and reduced the total number of modules in **U** (Supplementary Fig. 1d, e, h, i). Likewise, reducing the error for calculating **W** (i.e., smaller ldaW or larger $k$) also produced sparser **U**, likely because denser **W** allows sparser **U** within the same error tolerance in total (Supplementary Fig. 1d–k). Separating cells in **X** into blocks based on the vector size when calculating **W** (specified by Num_blocks_W) slightly reduced the sparsity of **U** (Supplementary Fig. 1h), but the effect size was minimal. SMAF produced slightly different **U** values over experimental replicates; however, the difference was minimal when the Lasso error tolerance coefficients (i.e., ldaW and ldaU) were not overly stringent (Supplementary Fig. 2). In summary, SMAF can be applied to protein expression data, and dictionaries with various levels of sparsity and decomposition accuracy can be calculated by adjusting the parameters in OMP or Lasso.

## CISI-IMC simulation protocol

We tested different dictionaries ($\mathbf{U} \in \mathbb{R}^{p \times d}$) and barcoding matrices ($\mathbf{A} \in \mathbb{R}^{m \times p}$) by simulating the CISI-IMC workflow using single-cell protein expression data ($\mathbf{X} \in \mathbb{R}^{p \times n}$) and evaluating the decompression performance, where $p$ is the number of proteins, $d$ is the number of modules in the dictionary, $m$ is the number of composite channels, and $n$ is the number of cells. CISI-IMC was simulated using the following steps: (1) Simulate single-cell composite data ($\mathbf{Y} \in \mathbb{R}^{m \times n}$) by compressing **X** using the barcoding matrix **A** by simple multiplication of $\mathbf{Y} = \mathbf{AX}$. (2) Calculate a sparse solution for **W** using Lasso. Note that the simulated composite data **Y** can be decomposed into $\mathbf{Y} = \mathbf{AUW}$, since $\mathbf{Y} = \mathbf{AX}$ and $\mathbf{X} = \mathbf{UW}$ according to the equations from compression and SMAF. As **A** and **U** are known, Lasso calculates the sparsest **W** within the error tolerance defined by $lda (=\lambda)$ where $\min_W \|\mathbf{W}\|_1 s.t. \|\mathbf{Y} - \mathbf{AUW}\|_2^2 < \lambda_W \|\mathbf{Y}\|_2^2$. (3) Reconstruct decompressed single-cell protein expression data $\widehat{\mathbf{X}}$ by calculation of $\widehat{\mathbf{X}} = \mathbf{UW}$. (4) Evaluate the decompression accuracy by calculating the correlation between the original **X** and the decompressed $\widehat{\mathbf{X}}$. Pearson's correlation coefficients were calculated separately for each protein, and the mean and minimum of the correlations, denoted as mean protein correlations and minimum protein correlations, respectively, were typically used for evaluation. The means of correlations calculated separately for each cell were also used and are referred to as mean cell correlations.

## Normalization of single-cell protein expression data to stimulate antibody titration

When simulating the antibody titration, we normalized **X** with weight wt, denoted as $\widetilde{\mathbf{X}}_{wt}$, before simulating composite data. When weight is 1, protein-wise L2-normalization was performed (i.e., each protein $x_p$ in $\widetilde{\mathbf{X}}_1$ has the same vector size). When $0 < wt < 1$, the weighted mean of **X** and $\widetilde{\mathbf{X}}_1$ was calculated using the following equation: $\widetilde{\mathbf{X}}_{wt} = wt\widetilde{\mathbf{X}}_1 + (1 - wt)\mathbf{X}$.

## Calculation of the dictionary

Twelve SMAF parameter conditions were tested on the training dataset according to the CISI-IMC simulation protocol. For each condition, 200 randomly generated barcoding matrices were used. For general assessment of correlations between the original **X** and the decompressed $\widehat{\mathbf{X}}$, 25% of the training dataset was used for simulating CISI-IMC and the rest of the training dataset was used for SMAF dictionary calculation. The same strategy was used for assessing the performance on original **X** with weighted normalization. For testing the performance stability in different tissues in the training dataset, the CISI-IMC simulation was separately performed on data from each tissue, and the dataset excluding the tissue was used for SMAF dictionary calculation. Correlations between the original **X** and the decompressed $\widehat{\mathbf{X}}$ were calculated for each tissue and were averaged across tissues. Final SMAF parameters were selected based on the overall performance of general assessment, stability across tissues, and stability across different normalization weight for **X**. The finalized dictionary was obtained with the selected SMAF parameters (algorithm_for_W : Lasso, ldaU : 0.02, ldaW : 0.02, Num_blocks_W : 1) calculated on the entire training dataset.

## Calculation of the barcoding matrix

To finalize the barcoding matrix **A**, random **A**s barcoding 16 proteins in 8 composite channels were generated with the restriction that each protein maximally used 2 composite channels. The selection process was separated into two rounds. In the first round, decompression performance of 2000 randomly generated **A**s was evaluated on 25% of the training data according to the CISI-IMC simulation protocol. **U** was calculated using the finalized SMAF parameters on the rest of the training data. The simulation was performed four times, and the top 50 **A**s were selected from each simulation based on the minimum protein correlations, which resulted in 200 **A** candidates from 8000 **A**s tested. For the second round, another four simulations were performed, except that the 200 **A** candidates were fixed for each. Minimum protein correlations for each **A** were averaged across the simulations for the selection of the best **A**.

## CISI-IMC data acquisition and decompression using the evaluation dataset

The barcoded antibody mix for CISI-IMC was combined with the ground-truth antibody mix for the evaluation dataset. Co-staining with the ground-truth antibody mix provided the matched ground-truth **X** for comparison with the decompressed **X**. Evaluation of actual CISI-IMC was performed as follows: (1) IMC data were obtained from tissues co-stained with the barcoded antibody mix and the ground-truth antibody mix. (2) IMC data analysis and single-cell segmentation yielded composite data **Y** and matched ground-truth **X**. (3) The sparse solution for **W** from **Y** was calculated using Lasso. Using known **A** and **U**, the sparsest **W** within the error tolerance defined by $lda (=\lambda)$ where $\min_W \|\mathbf{W}\|_1 s.t. \|\mathbf{Y} - \mathbf{AUW}\|_2^2 < \lambda \|\mathbf{Y}\|_2^2$, was determined using Lasso. (4) Decompressed single-cell protein expression data $\widehat{\mathbf{X}}$ was reconstructed ($\widehat{\mathbf{X}} = \mathbf{UW}$). (5) Decompression accuracy was evaluated by comparing ground-truth **X** and decompressed $\widehat{\mathbf{X}}$.

## Cell-type classification with random forest classifier

To classify cells into cell types, we used the random forest classifier implemented in the caret package[17]. Training data was prepared by selecting cells with known marker expression for each cell type. To account for the imbalanced frequency of cell types in the training data, the Random Over-Sampling Examples sampling method was used[18]. Two classifiers were trained using ground-truth data or decompressed data separately. After predicting the cell types of all cells based on ground-truth expression using the classifier trained on ground-truth data, cells with low probability to be any cell type was treated as undefined and were removed from further analyses since the cells were likely to be tumor cells or other cells that have low level of expression of measured markers. Cells assigned with a cell type using ground-truth data were also classified using decompressed expression with the classifier trained on decompressed data. Obtained cell types

from ground-truth data and decompressed data were compared to evaluate the accuracy of CISI-IMC decompression.

## Reporting summary

Further information on research design is available in the Nature Portfolio Reporting Summary linked to this article.

## Data availability

Tiff files for IMC data and single-cell data generated in this study have been deposited in Zenodo (https://doi.org/10.5281/zenodo.17533580). Source data are provided with this paper.

## Code availability

All code used for preprocessing IMC data, SMAF dictionary calculation, and decompressing composite data is available at https://github.com/BodenmillerGroup/CISI-IMC.

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

## Acknowledgements

We thank all the member of B.B. laboratory. We especially appreciate the assistance of N. de Souza with writing of the manuscript. B.B. was funded by two SNSF project grants (#310030_205007: Analysis of breast tumor ecosystem properties for precision medicine approaches and #316030_213512: Cellular-resolution high-performance mass spectrometric), an NIH grant (cruk), the CRUK IMAXT Grand Challenge, and the European Research Council (ERC) under the European Union's Horizon 2020 Program under the ERC grant agreement no. 866074 ("Precision Motifs").

## Author contributions

T.H. and B.B. conceived the study. T.H. performed IMC experiments. T.H., L.S.S. and N.E. performed computational analyses. H.M. produced microarray of tumor and healthy tissues. T.H. and B.B. wrote the manuscript.

## Competing interests

B.B. is a founder and shareholder of Navignostics, a precision oncology spin-off from the University of Zurich, based on multiplexed imaging. All other authors declare no competing interests.
