## [Transparent Peer Review file · Nature Communications]

Compressed sensing expands the multiplexity of imaging mass cytometry

Corresponding Author: Professor Bernd Bodenmiller

Version 0:

Reviewer comments:

Reviewer #1

(Remarks to the Author)

The authors have addressed my comments and highlighted in the revised version. However, since this was trained and tested on immune/stromal cells, I think this should be more clearly highlighted in the abstract, similar to the statement in the discussion:

"The expression dictionary and barcoding matrix we describe here is therefore immediately implementable by IMC users for general immune and stromal cell type classification."

(Remarks on code availability)

Reviewer #3

(Remarks to the Author)

I will here number and refer directly to the authors' responses to my previous comments.

Point 1 (Effect of mathematical choices on biological interpretation)

I would still argue that exactly the endpoint of the final application should be what the performance should be assessed for and what should be ultimately optimized for? If it does skew the analysis, even in its current implementation, that is important information to have.

Point 2 (Influence of low probability cell types on performance)

I don't think the authors have sufficiently addressed my initial comment. It is important to know how the performance of this approach is in a real world scenario with ambiguous and hard to classify cell types present.

Point 3 (Lower performance on small subpopulations)

Adjusting the text to reflect the fact that it is not reliable for marker expression is fine. For lower frequency cell types, I still do wonder how the performance would be influenced if for example tumor cells would have been included here which could make up easily up to 75% of all cells. Would that have affected performance for other cell types, since there seems to be a (slight) correlation with their frequency (which is not reported though).

Point 4 (Adding more proteins)

This might be true for the mathematical error but is not particularly reassuring for researchers interested in particular cell types. Overall, from the responses, I do not have a good understanding of the expandability of the approach.

Point 5 (False annotation)

Despite other problems in spatial proteomics, this approach would still introduce additional ones, overall limiting applicability in my opinion.

Point 6 (related to 4)
OK, see point 4

Point 7 (Pixel-level)
OK

Point 8 (Other technologies)
OK, but limits the broader applicability

(Remarks on code availability)

REVIEWER COMMENTS

Reviewer #1 (Remarks to the Author):

The authors have addressed my comments and highlighted in the revised version. However, since this was trained and tested on immune/stromal cells, I think this should be more clearly highlighted in the abstract, similar to the statement in the discussion:

"The expression dictionary and barcoding matrix we describe here is therefore immediately implementable by IMC users for general immune and stromal cell type classification."

We have modified the abstract accordingly.

Reviewer #3 (Remarks to the Author):

I will here number and refer directly to the the authors reponses to my previous comments.

Point 1 (Effect of mathematical choices on biological interpretation)

I would still argue that exactly the endpoint of the final application should be what the performance should be assessed for and what should be ultimately optimized for? If it does skew the analysis, even in its current implementation, that is important information to have.

We accept the reviewer's point that, compared to marker-level correlation, cell type classification accuracy is more biologically interpretable. However, it is also more sensitive to many factors other than decompression accuracy itself, such as how each marker contributes to the classification of each cell type, the level of cell type granularity, and the uncertainty threshold to call a cell as a given cell type. In contrast, marker-level correlation directly reflects the decompression accuracy, which is what we wanted to compare across different parameters. This is why we used marker-level correlation for optimizing parameters for SMAF and the barcoding matrix. We note that the previous CISI paper (Cleary et. al., 2021) and a recent paper using similar decompression approach called CombPlex (Ben-Uri et. al., 2025) also use marker-level correlation (and marker-level F1 score for CombPlex) to optimize algorithm parameters. Altogether, we believe marker-level correlation is the best proxy for decompression performance when optimizing SMAF parameters.

In the current manuscript, we show that the choice of SMAF parameters affects the marker correlation to a level of ~10% (Supplementary Fig, 5). While additional calculation of the cell type classification accuracy for each SMAF parameter condition is theoretically possible (although time-consuming as it is computationally very demanding), in our view, the result will be difficult to interpret in terms of decompression performance due to the many factors that could affect the classification accuracy as discussed above.

Point 2 (Influence of low probability cell types on performance)

I dont think the authors have sufficiently addressed my initial comment. It is important to know how the performance of this approach is in a real world scenario with ambiguous and hard to classify cell types present.

We acknowledge that the data in this manuscript do not perfectly reflect a real-world scenario because we removed cells that had low probability of being any cell type from the analysis (using ground truth data). However, repeating the analysis with the removed cells now added back and labeled as “ambiguous” would not usefully represent a real-world scenario either, as they are the majority of the cells in the validation dataset. This is because they are likely tumor cells and we could not annotate them because we did not include tumor markers in the antibody panel. Addressing the referee’s point would require repeating the study with new marker panels, including a pan-tumor marker, so that a small fraction of uncertain/ambiguous cells could be included. We consider this out of the scope of the current manuscript. We now more clearly discuss this limitation of our analysis (lines 270-272). Also, to mitigate the issue of ambiguous cells, we now explicitly suggest that users include general cell-type markers in their antibody panels to minimize the number of cells that have little signal from any marker (lines 272-274).

Point 3 (Lower performance on small subpopulations)

Adjusting the text to reflect the fact that it is not reliable for marker expression is fine. For lower frequency cell types, I still do wonder how the performance would be influenced if for example tumor cells would have been included here which could make up easily up to 75% of all cells. Would that have affected performance for other cell types, since there seems to be a (slight) correlation with their frequency (which is not reported though).

The referee asks about low frequency cell types. As also shown in the previous CISI publication by Cleary et., al, markers with low frequency and low intensity were decompressed relatively poorly, as we stated (lines 248-251). Further, our simulations show that decompression performance improves when normalizing marker expression for both marker frequency and intensity (Supplementary Fig. 4). It is therefore expected that if a marker for high-frequency cells such as tumor cells is added to the current CISI antibody panel, the performance for other, less frequent cells may be reduced. We added a line recommending that, if the CISI panel includes a marker that is expected to be expressed by a much higher number of cells than other markers, titrating this marker to a lower signal intensity than other markers may improve the overall decompression accuracy (line 257-259).

Point 4 (Adding more proteins)

This might be true for the mathematical error but is not particularly reassuring for researchers interested in particular cell types. Overall, from the responses, I do not have a good understanding of the expandability of the approach.

Our claim is that CISI-IMC allows the measurement of more markers per channel, but we acknowledge that this introduces error. Performance with an expanded panel is likely to depend on the markers and cell types added, we find it difficult to make general predictions. However, to aid in implementation of the approach, we now explicitly say that markers for which quantification is critical should be added outside the compressed channel (line 275), that antibodies detecting compressed markers should be titrated to a similar intensity for optimal decompression performance (line 256), and that the panel (compressed and non-compressed) should be designed such that there are not many cells that are not stained with any marker (line 272).

Point 5 (False annotation)

Despite other problems in spatial proteomics, this approach would still introduce additional ones,

overall limiting applicability in my opinion.

Please see the response to point 4 above. The reviewer is of course correct that almost all gains in performance in some parameters of an imaging method come with concomitant losses in other parameters. We hope that the manuscript is now more transparent about these limitations.

Point 6 (related to 4)

OK, see point 4

Point 7 (Pixel-level)

OK

Point 8 (Other technologies)

OK, but limits the broader applicability